# Self-reported eating rate and metabolic syndrome in Japanese people: cross-sectional study

Satsue Nagahama,[1,2,3] Kayo Kurotani,[1] Ngoc Minh Pham,[4] Akiko Nanri,[1] Keisuke Kuwahara,[1] Masashi Dan,[2] Yuji Nishiwaki,[3] Tetsuya Mizoue[1]

[1]Department of Epidemiology and Prevention, Center for Clinical Sciences, National Center for Global Health and Medicine, Shinjuku-ku, Tokyo, Japan
[2]All Japan Labor Welfare Foundation, Tokyo, Japan
[3]Department of Environmental and Occupational Health, School of Medicine, Toho University, Tokyo, Japan
[4]Department of Epidemiology, Faculty of Public Health, Thai Nguyen University of Medicine and Pharmacy, Thai Nguyen Province, Vietnam

**Correspondence to**
Dr Satsue Nagahama;
satsue_0323@hotmail.com

## ABSTRACT

**Objectives:** To examine the association between self-reported eating rate and metabolic syndrome.

**Design:** Cross-sectional study.

**Setting:** Annual health checkup at a health check service centre in Japan.

**Participants:** A total of 56 865 participants (41 820 male and 15 045 female) who attended a health checkup in 2011 and reported no history of coronary heart disease or stroke.

**Main outcome measure:** Metabolic syndrome was defined by the joint of interim statement of the International Diabetes Federation and the American Heart Association/National Heart, Lung, and Blood Institute.

**Results:** In multiple logistic regression models, eating rate was significantly and positively associated with metabolic syndrome. The multivariable-adjusted ORs (95% CI) for slow, normal and fast were 0.70 (0.62 to 0.79), 1.00 (reference) and 1.61 (1.53 to 1.70), respectively, in men (p for trend <0.001), and 0.74 (0.60 to 0.91), 1.00 (reference) and 1.27 (1.13 to 1.43), respectively, in women (p for trend <0.001). Of metabolic syndrome components, abdominal obesity showed the strongest association with eating rate. The associations of eating rate and metabolic syndrome and its components were largely attenuated after further adjustment for body mass index; however, the association of slow eating with lower odds of high blood pressure (men and women) and hyperglycaemia (men) and that of fast eating with higher odds of lipid abnormality (men) remained statistically significant.

**Conclusions:** Results suggest that eating rate is associated with the presence of metabolic syndrome and that this association is largely accounted for by the difference in body mass according to eating rate.

## INTRODUCTION

Metabolic syndrome (MetS) is a cluster of physiological risk factors associated with cardiovascular disease and several types of cancer.[1] Determination of aetiological factors for MetS is required for the establishment of public health strategies to reduce its

### Strengths and limitations of this study

- This study included a large number of participants, used waist circumference in defining metabolic syndrome and analysed data for men and women separately.
- Eating rate was assessed by a self-reported questionnaire.
- Information on dietary intake was not obtained.

prevalence and prevent resulting complications. Growing evidence from both observational and interventional studies suggests that dietary habits have a role in the development of MetS,[2–4] which originates from obesity. Obesity has been extensively investigated in relation to dietary habits including eating rate since 1962, when Ferster *et al* published a theoretical and practical weight control programme focusing on eating behaviours including eating rate.[5] Observational studies showed that obese people ate at a faster rate than non-obese people,[6] and reducing the eating rate may be a simple and effective treatment for obesity.[7]

During the past decade, several cross-sectional studies have found a positive association between eating rate and overweight[8–11] or insulin resistance.[11–15] Similarly, a few longitudinal studies have shown that eating quickly is associated with an increased risk of weight gain[16 17] and type 2 diabetes.[18] In addition, some cross-sectional studies have reported that fast eating is positively associated with hypertriglyceridaemia and low high-density lipoprotein cholesterol (HDL-C).[11 14 19] Therefore it is conceivable that eating rate might be associated with MetS. To our knowledge, however, only one Korean study cross-sectionally examined eating rate in relation to MetS.[20] In that study, MetS was defined using body mass index (BMI), rather than waist circumference, and investigated the association in men only. Waist circumference is a component of most MetS

definitions as a surrogate for central obesity, which can better predict cardiovascular risk.[21] It is therefore necessary to examine the relationship between eating rate and MetS using waist circumference in both men and women. Here, we investigated cross-sectionally the association between self-reported eating rate and the presence of MetS according to the joint interim statement (JIS) of the International Diabetes Federation and the American Heart Association/National Heart, Lung, and Blood Institute[22] using a large dataset from health checkups in Japanese men and women.

## METHODS AND PROCEDURES
### Study population

In Japan, a health checkup under occupational health and safety law is mandatory for all employees.[23] The law was modified in 2008 when a recommendation for a new national health checkup system focusing on MetS was launched.[24] Study participants attended the 2011 (calendar year) annual health checkup at the All Japan Labor Welfare Foundation (Tokyo)—a health service centre. Participants were mainly Japanese employees but also included a small number of their dependants and foreign workers—men aged 17–99 and women aged 17–85. Of 297 148 participants, we excluded 3660 with a history of myocardial infarction, coronary heart disease or stroke, which might influence both eating rate and MetS. Of the remaining 293 488 participants, we included 269 297 who reported their eating rate. Of these, we excluded 182 487 with missing data for any of the components of MetS (173 376 without plasma glucose, 61 602 without waist circumference, 43 724 without triglyceride, 43 401 without HDL-C and 504 without blood pressure; some participants had two or more missing data). The main reason for the large number of participants with missing blood glucose measurement was that HbA1c was measured instead of blood glucose for those who attended the checkup in a non-fasting condition. Of the remaining 86 810 participants, we further excluded 29 337 who had a meal within 8 h before blood was drawn or provided no information about meal time. After further exclusion of 608 participants with missing information on covariates (BMI, smoking status, alcohol consumption and physical activity), 56 865 participants (41 820 male and 15 045 female) remained for analysis.

We did not obtain written informed consent from each participant; instead, we disclosed our intention to carry out the study by putting up posters, giving participants an opportunity to refuse the use of their data for the study. In Japan, informed consent is not necessarily required for observational studies using existing data, as described in the Japanese Ethical Guidelines for Epidemiological Research.[25] The research protocol was approved by the ethics committee of the National Center for Global Health and Medicine and the ethics committee of Toho University.

### Data collection and measurements

A self-administered questionnaire, which was recommended for specific health examination by the Japanese government (Ministry of Health, Labour and Welfare),[26] was used to assess eating rate, medical history and health-related lifestyles, including smoking, alcohol consumption and regular physical activity. Eating rate was assessed by asking "How fast is your speed of eating?", with three response options (slow, normal and fast). A trained staff measured height to the nearest 0.1 cm, weight to the nearest 0.1 kg and waist circumference to the nearest 0.1 cm at the umbilical level in a standing position. BMI was calculated as the weight in kilograms divided by the squared height in metres. Blood pressure in the sitting position was measured using an automated machine (HEM-907, Omron, Kyoto, Japan). Participants with high blood pressure (≥130 mm Hg systolic or ≥85 mm Hg diastolic) received another measurement and data showing the lower systolic blood pressure were used. A venous blood sample was collected, stored in a cooler at 4°C for transportation to an external laboratory (SRL, Tokyo, Japan) and measured within 24 h of being drawn. Triglyceride level was measured by an enzymatic colorimetric test (Bio Majesty JCA-BM8060, JEOL, Tokyo, Japan) and HDL-C was determined by a direct method (Bio Majesty JCA-BM8060). Plasma glucose level was determined by the hexokinase method (an automatic clinical chemistry analyser JCA-BM9000 series, JEOL, Tokyo, Japan).

### Definitions for MetS

According to the JIS criteria, MetS was defined as three or more of the following risk factors: (1) waist circumference for Asian population ≥90 cm in men and ≥80 cm in women, (2) triglyceride level ≥150 mg/dL (1.7 mmol/L), (3) HDL-C level <40 mg/dL (1.04 mmol/L) in men and <50 mg/dL (1.3 mmol/L) in women, (4) blood pressure ≥130 mm Hg systolic or ≥85 mm Hg diastolic, (5) fasting glucose level ≥100 mg/dL (5.6 mmol/L).[22] Participants receiving medication for diabetes, hypertension and dyslipidaemia were considered as having the respective factor, irrespective of measured data.

### Statistical analysis

Participants were divided into three groups according to their eating rate (slow, normal and fast). The characteristics of participants for each eating rate category were expressed as means (SD) for continuous variables and percentages for categorical variables, respectively. Fasting plasma glucose and triglyceride were highly skewed; hence, they were log-transformed and expressed as geometric means (95% CIs). Trend association was assessed by assigning ordinal numbers (0–2) to the categories of eating rate (slow, normal and fast, respectively) and was tested using linear regression and logistic regression, as appropriate. Multiple logistic regression analysis was used to estimate the ORs with 95% CI for the presence of MetS across eating rate categories, with normal eating

rate as the reference. We adjusted for age (continuous, year) in the basic model. In the second model, we further adjusted for smoking status (non-smoker, daily smoker consuming <20 cigarettes a day or ≥20 cigarettes a day), physical activity (walking time <60 min a day or ≥60 min a day) and alcohol consumption (non-drinker, <1 *go*, 1 to <2 *go* or ≥2 *go* per day; one *go* of sake, Japanese traditional beverage, is about 180 mL of 10–14% of ethanol and contains ~23 g of ethanol). In the third model, we added BMI (continuous, kg/m$^2$) to the second model. We performed a likelihood ratio test for the interaction between eating rate and sex. All analyses were done for men and women separately because the interaction was significant (p for interaction <0.001). We repeated the above analyses for each component of MetS. A two-tailed p value <0.05 was considered statistically significant. All statistical analyses were performed with STATA, V.12.1 (StataCorp, College Station, Texas, USA).

## RESULTS

The prevalence of MetS was 18.5% in men and 12.8% in women. Table 1 shows characteristics of the study participants across categories of eating rate. Men who ate quickly tended to be young, whereas women who ate slowly tended to be young. Both men and women who ate quickly consumed a greater amount of alcohol and had significantly higher BMI, waist circumference, triglyceride level and systolic and diastolic blood pressures and lower HDL-C level than those who ate slowly.

The ORs of the presence of MetS across eating rate are shown in table 2. Faster eating was associated with a higher presence of MetS in age- and multivariable-adjusted models. The trend was more apparent in men than in women. The multivariable-adjusted ORs (95% CI) of MetS for slow, normal and fast rate eating were 0.70 (0.62 to 0.79), 1.00 (reference) and 1.61 (1.53 to 1.70), respectively, in men (p for trend <0.001), and 0.74 (0.60 to 0.91), 1.00 (reference) and 1.27 (1.13 to 1.43), respectively, in women (p for trend <0.001). Further adjustment for BMI markedly attenuated these associations; however, the association with fast eating and MetS remained statistically significant in men.

Table 3 shows the ORs of the presence of individual MetS components across three categories of eating rate. Central obesity sharply increased with increasing speed of eating; the ORs for slow, normal and fast eating were 0.63, 1.00 (reference) and 1.97, respectively, in men (p for trend <0.001), and 0.73, 1.00 (reference) and 1.44, respectively, in women (p for trend <0.001). High blood pressure and high triglyceride were positively associated with eating rate in both men and women. High fasting plasma glucose and low HDL-C were associated with fast eating in both men and women, but they were associated with slow eating in men only. Additional adjustment for BMI largely attenuated these associations and the significant trend association disappeared. However, the associations of slow eating with decreased odds of high blood pressure (men and women) and hyperglycaemia (men) and those of fast eating with increased odds of abnormal lipid profile (men) remained statistically significant.

## DISCUSSION

In this large population of Japanese men and women, we found that eating rate was positively associated with the presence of MetS, especially in men. Of the components of MetS, the association with abdominal obesity was strongest. These associations were largely attenuated after adjustment for BMI. However, slow eating was associated with a decreased odds of high blood pressure in both men and women and high fasting plasma glucose in men, and fast eating was associated with increased odds of abnormal lipid profiles in men. To the best of our knowledge, this study is the first to report a positive association between eating rate and MetS defined by using waist circumference.

This finding for MetS is consistent with that of a study among Korean men reporting that eating rate was positively associated with MetS, which was defined using BMI instead of waist circumference.[20] For the MetS components, our study is compatible with some cross-sectional studies showing that eating rate is associated with a higher BMI,[8–11] and two longitudinal studies showing that eating rate is associated with weight gain.[16 17] In a Korean study that determined the association between eating rate and components of MetS for men and women separately, eating rate was associated with obesity, high blood pressure, hyperglycaemia and abnormal lipid profile in men, whereas it was associated with only obesity in women.[11] Our results were largely consistent with those in the Korean study (except for blood pressure in women).

Notably, we found that the associations of MetS components with eating rate were largely attenuated after adjustment for BMI, a finding compatible with those of a cross-sectional study in Korea[11] and a prospective study in Japan.[18] This result indicates that obesity is a mediator whereby fast eating causes MetS components to deteriorate. We also found, however, that some associations remained statistically significant even after adjusting for BMI (dyslipidaemia with fast eating and hyperglycaemia with slow eating in men, and high blood pressure with slow eating in both men and women). Similarly, the above-mentioned Korean study[11] reported that a fast rate of eating remained an important determinant for low HDL-C and high fasting plasma glucose after adjustment for BMI in men. Therefore, there may be pathways other than weight gain that underlie the association between eating rate and MetS.

We found that the association between eating rate and MetS was stronger in men than in women, consistent with a previous study in Korea.[11] Such sex difference may reflect the difference in eating speed of men and women. One study found that women took more bites, had a smaller bite size and slower bites than men in

**Table 1** Characteristics of the study individuals according to eating rates

| Characteristics | Men (n=41 820) | | | p For trend* | Women (n = 15 045) | | | p For trend* |
|---|---|---|---|---|---|---|---|---|
| | Slow | Normal | Fast | | Slow | Normal | Fast | |
| n (%) | 2821 (6.8) | 24 893 (59.5) | 14 106 (33.7) | | 1398 (9.3) | 9893 (65.8) | 3754 (24.9) | |
| Age (years)† | 46.9±12.3 | 46.9±10.9 | 45.0±10.4 | <0.001 | 43.5±12.5 | 47.2±11.6 | 46.7±11.2 | <0.001 |
| Walking time, ≥60 min/day (%) | 21.8 | 19.0 | 20.6 | 0.004 | 15.5 | 15.0 | 16.1 | 0.798 |
| Smoking status (%) | | | | | | | | |
| Non-smoker | 61.9 | 55.0 | 56.6 | <0.001 | 82.9 | 83.1 | 80.7 | 0.572 |
| Daily consuming <20 cigarettes/day | 28.6 | 34.6 | 31.3 | | 16.0 | 15.7 | 17.6 | |
| Daily consuming ≥20 cigarettes/day | 9.5 | 10.4 | 12.1 | | 1.1 | 1.2 | 1.7 | |
| Alcohol (%) | | | | | | | | |
| Non-drinker | 30.2 | 26.7 | 26.7 | <0.001 | 53.4 | 52.2 | 49.8 | <0.001 |
| Drinker <1 *go*/day‡ | 33.9 | 35.7 | 34.5 | | 34.2 | 35.9 | 35.5 | |
| Drinker 1 to <2 *go*/day‡ | 24.6 | 26.3 | 26.8 | | 9.5 | 9.5 | 11.5 | |
| Drinker ≥2 *go*/day‡ | 11.3 | 11.3 | 12.0 | | 2.9 | 2.4 | 3.2 | |
| BMI (kg/m$^2$)† | 22.4±3.3 | 23.4±3.3 | 24.6±3.7 | <0.001 | 21.0±3.5 | 21.8±3.5 | 22.5±3.8 | <0.001 |
| Waist circumference (cm)† | 80.3±9.2 | 82.9±9.0 | 86.0±9.8 | <0.001 | 75.5±9.5 | 77.7±9.4 | 79.6±9.8 | <0.001 |
| Systolic blood pressure (mm Hg)† | 123.5±15.5 | 126.1±15.5 | 126.7±15.1 | <0.001 | 113.1±16.3 | 117.3±17.2 | 117.0±17.2 | <0.001 |
| Diastolic blood pressure (mm Hg)† | 75.2±11.4 | 77.3±11.9 | 78.0±12.0 | <0.001 | 69.1±10.9 | 71.4±11.5 | 71.5±11.9 | <0.001 |
| Fasting plasma glucose (mg/dL)§ | 93.0 (92.5 to 93.6) | 94.4 (94.2 to 94.6) | 94.6 (94.3 to 94.8) | <0.001 | 88.1 (87.5 to 88.7) | 89.1 (88.9 to 89.3) | 89.5 (89.1 to 89.9) | <0.001 |
| Triglyceride (mg/dL)§ | 98.3 (96.2 to 100.4) | 103.8 (103.0 to 104.6) | 110.8 (109.8 to 111.9) | <0.001 | 67.0 (65.3 to 68.7) | 71.6 (70.9 to 72.3) | 74.1 (72.9 to 75.2) | <0.001 |
| HDL-C (mg/dL)† | 61.3±15.3 | 59.4±15.0 | 57.2±14.3 | <0.001 | 71.4±15.3 | 70.5±15.8 | 69.3±15.5 | <0.001 |

Cross-sectional survey of 56 865 examinees in All Japan Labor Welfare Foundation, Japan, 2011.
*Linear regression and logistic regression, assigning ordinal number (0 to 2) to eating rate, as appropriate.
†Mean±SD.
‡One *go* contains ~23 g of ethanol.
§Geometric means (95% CIs).
BMI, body mass index; HDL-C, high-density lipoprotein cholesterol.

**Table 2** ORs and 95% CIs for metabolic syndrome according to eating rate (n=56 865)

| Eating rate | Men (n=41 820) | | | | Women (n=15 045) | | | |
|---|---|---|---|---|---|---|---|---|
| | Slow | Normal* | Fast | p For trend† | Slow | Normal* | Fast | p For trend† |
| n (%) | 2821 (6.8) | 24 893 (59.5) | 14 106 (33.7) | | 1398 (9.3) | 9893 (65.8) | 3754 (24.9) | |
| MetS, n | 361 | 4180 | 3193 | | 116 | 1261 | 547 | |
| Model 1 ‡ | 0.70 (0.62 to 0.79) | 1.00 | 1.62 (1.53 to 1.71) | <0.001 | 0.75 (0.61 to 0.92) | 1.00 | 1.27 (1.13 to 1.42) | <0.001 |
| Model 2 § | 0.70 (0.62 to 0.79) | 1.00 | 1.61 (1.53 to 1.70) | <0.001 | 0.74 (0.60 to 0.91) | 1.00 | 1.27 (1.13 to 1.43) | <0.001 |
| Model 3 ¶ | 0.91 (0.80 to 1.04) | 1.00 | 1.10 (1.03 to 1.17) | <0.001 | 0.88 (0.70 to 1.11) | 1.00 | 0.98 (0.86 to 1.12) | 0.714 |

MetS as defined using the criteria of the joint interim statement: the presence of three or more of the following risk factors: (1) waist circumference ≥90 cm in men and ≥80 cm in women, (2) triglyceride level ≥150 mg/dL (1.7 mmol/L), (3) HDL-C level <40 mg/dL (1.04 mmol/L) in men and <50 mg/dL (1.3 mmol/L) in women, (4) blood pressure ≥130 mm Hg systolic or ≥85 mm Hg diastolic, (5) fasting glucose level ≥100 mg/dL (5.6 mmol/L).
*Reference.
†Multiple logistic regression, assigning ordinal number (0–2) to eating rate.
‡Adjusted for age.
§Adjusted for age, smoking status, alcohol and regular physical activity.
¶Adjusted for age, smoking status, alcohol, regular physical activity and body mass index.
MetS, Metabolic syndrome.

eating the same amount of doughnut, irrespective of body size.[27] Another study showed that objectively measured eating speed in men with self-reported slow eating was faster than that in women with self-reported fast eating.[28] Taken together, eating rate may have a greater impact on metabolism in men than that in women.

Although mechanisms by which eating rate influences metabolism have not been fully elucidated, overeating may link fast eating to MetS. Fast eating passes few satiety signals from the mouth to the brain,[29 30] induces less satiation owing to a lack of stomach expansion[31] and alters the circulating levels of certain gut hormones.[32 33] In these pathways, fast eating leads to excess energy intake,[34 35] resulting in overweight and MetS. Because fast eating has been associated with obesity even after adjusting for total energy intake,[8–11 14] other pathways may operate. One study showed that interleukin-1β and interleukin-6 levels were higher among those who ate quickly than among those who ate slowly, even after accounting for energy intake and BMI.[36] These cytokines could induce insulin resistance,[37 38] contributing to high blood pressure via an increased renal sodium and water retention, plasma noradrenaline and sympathetic nervous system activity.[39–41]

The strengths of our study should be mentioned. This study has a large sample size (56 865 participants). In addition, body weight, body height and waist circumference were measured by trained technicians, which increased the validity of our study. Nonetheless, the study has several limitations. First, eating rate was self-reported. However, a self-reported eating rate has been shown to be well correlated with that reported by a friend[9] or one that is objectively measured.[28] Second, information on dietary intake was not available and thus total energy intake was not considered in the analyses. The adjustment of energy intake, however, might not be appropriate because energy intake may increase with eating rate and thus might act as a mediator rather than confounder. Moreover, eating rate has been associated with body weight independently of energy intake.[7–10 13] Third, fast-food is an energy-dense dietary source and has been linked to MetS.[42] Because fast-food is usually consumed quickly, it may confound the association of eating rate with MetS. Fourth, the study participants were mainly workers in various industries including manufacturing (43.6%), service (27.8%) and transport and telecommunications (9.9%), and these figures are similar to those of national survey.[43] However, information on the profession of participants was not available, and thus caution is required when generalising our findings. Fifth, a large number of participants were excluded from the analysis owing to missing data for MetS components. We cannot deny the possibility of bias due to such selective inclusion. Sixth, the cross-sectional design precludes any causal inferences about the role of eating rate. Finally, we cannot exclude the possibility of residual confounding and confounding by unmeasured variables.

**Table 3** ORs and 95% CIs for components of metabolic syndrome according to eating rate (n=56 865)

| Eating rate | Men (n=41 820) | | | | Women (n = 15 045) | | | |
|---|---|---|---|---|---|---|---|---|
| | Slow | Normal* | Fast | p For trend† | Slow | Normal* | Fast | p For trend† |
| n (%) | 2821 (6.8) | 24 893 (59.5) | 14 106 (33.7) | | 1398 (9.3) | 9893 (65.8) | 3754 (24.9) | |
| Central obesity‡ | | | | | | | | |
| Model 1§ | 0.63 (0.56 to 0.71) | 1.00 | 1.98 (1.89 to 2.08) | <0.001 | 0.73 (0.64 to 0.83) | 1.00 | 1.44 (1.34 to 1.56) | <0.001 |
| Model 2¶ | 0.63 (0.56 to 0.70) | 1.00 | 1.97 (1.88 to 2.07) | <0.001 | 0.73 (0.64 to 0.83) | 1.00 | 1.44 (1.33 to 1.56) | <0.001 |
| High blood pressure** | | | | | | | | |
| Model 1§ | 0.75 (0.69 to 0.82) | 1.00 | 1.22 (1.17 to 1.27) | <0.001 | 0.76 (0.66 to 0.88) | 1.00 | 1.10 (1.01 to 1.21) | <0.001 |
| Model 2¶ | 0.74 (0.68 to 0.81) | 1.00 | 1.20 (1.15 to 1.26) | <0.001 | 0.76 (0.65 to 0.88) | 1.00 | 1.10 (1.00 to 1.20) | <0.001 |
| Model 3†† | 0.88 (0.81 to 0.96) | 1.00 | 0.97 (0.93 to 1.02) | 0.645 | 0.85 (0.72 to 0.99) | 1.00 | 0.93 (0.84 to 1.02) | 0.923 |
| High fasting plasma glucose‡‡ | | | | | | | | |
| Model 1§ | 0.78 (0.71 to 0.87) | 1.00 | 1.17 (1.12 to 1.23) | <0.001 | 1.03 (0.85 to 1.25) | 1.00 | 1.17 (1.04 to 1.31) | 0.035 |
| Model 2¶ | 0.78 (0.71 to 0.87) | 1.00 | 1.16 (1.11 to 1.22) | <0.001 | 1.03 (0.85 to 1.25) | 1.00 | 1.16 (1.03 to 1.31) | 0.042 |
| Model 3†† | 0.88 (0.80 to 0.98) | 1.00 | 0.99 (0.94 to 1.05) | 0.238 | 1.14 (0.94 to 1.40) | 1.00 | 1.02 (0.90 to 1.15) | 0.536 |
| High triglyceride§§ | | | | | | | | |
| Model 1§ | 0.88 (0.80 to 0.96) | 1.00 | 1.32 (1.26 to 1.38) | <0.001 | 0.83 (0.67 to 1.01) | 1.00 | 1.14 (1.01 to 1.28) | 0.002 |
| Model 2¶ | 0.90 (0.82 to 0.98) | 1.00 | 1.33 (1.27 to 1.39) | <0.001 | 0.81 (0.66 to 1.00) | 1.00 | 1.13 (1.01 to 1.27) | 0.002 |
| Model 3†† | 1.08 (0.98 to 1.19) | 1.00 | 1.07 (1.02 to 1.12) | 0.121 | 0.90 (0.73 to 1.11) | 1.00 | 0.98 (0.87 to 1.11) | 0.753 |
| Low HDL-C¶¶ | | | | | | | | |
| Model 1§ | 0.83 (0.73 to 0.96) | 1.00 | 1.34 (1.26 to 1.43) | <0.001 | 0.90 (0.74 to 1.09) | 1.00 | 1.11 (0.99 to 1.25) | 0.018 |
| Model 2¶ | 0.83 (0.73 to 0.96) | 1.00 | 1.36 (1.28 to 1.45) | <0.001 | 0.89 (0.74 to 1.08) | 1.00 | 1.12 (1.00 to 1.26) | 0.011 |
| Model 3†† | 0.97 (0.84 to 1.12) | 1.00 | 1.10 (1.03 to 1.18) | 0.004 | 1.00 (0.82 to 1.22) | 1.00 | 0.96 (0.85 to 1.08) | 0.500 |

*Reference.
†Multiple logistic regression, assigning ordinal number (0–2) to eating rate.
‡Waist circumference ≥90 cm in men and ≥80 cm in women.
§Adjusted for age.
¶Adjusted for age, smoking status, alcohol and regular physical activity.
**Blood pressure ≥130 mm Hg for systolic or ≥85 mm Hg for diastolic.
††Adjusted for age, smoking status, alcohol, regular physical activity and body mass index.
‡‡Fasting plasma glucose ≥100 mg/dL or under medication.
§§Triglyceride ≥150 mg/dL or receiving medication.
¶¶HDL-C <40 mg/dL in men, <50 mg/dL in women or receiving medication.
BMI, body mass index; HDL-C, high-density lipoprotein cholesterol.

In conclusion, we found a positive trend association between self-reported eating rate and the presence of MetS in men and women. The association between eating rate and MetS was largely accounted for by the difference in body mass across eating rate. Further research should examine whether reducing eating rate prevents obesity and MetS.

**Acknowledgements** The authors would like to thank Dr Nobuo Yanagisawa and Dr Takeshi Kawaguchi for coordinating the study.

**Contributors** SN and KaK designed study and drafted the manuscript. SN, NMP, AN, KeK performed the data analysis. MD collected and interpreted the data. All authors participated in interpretation of the findings, revised the paper critically for important intellectual content and approved the final version to be published. TM and YN provided administrative, technical and material support. SN and TM are guarantors.

**Funding** This study was supported by the Industrial Health Foundation.

**Competing interests** None.

**Ethics approval** The research protocol was approved by the ethics committee of the National Center for Global Health and Medicine and the ethics committee of Toho University.

**Provenance and peer review** Not commissioned; externally peer reviewed.

**Data sharing statement** No additional data are available.

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
