## [Reviewer comments · BMJ Open]

Some articles will have been accepted based in part or entirely on reviews undertaken for other BMJ Group journals. These will be reproduced where possible.

ARTICLE DETAILS

TITLE (PROVISIONAL)	Self-reported eating rate and metabolic syndrome in Japanese: cross-sectional study
AUTHORS	Nagahama, Satsue; Kurotani, Kayo; Pham, Ngoc Minh; Nanri, Akiko; Kuwahara, Keisuke; Dan, Masashi; Nishiwaki, Yuji; Mizoue, Tetsuya

VERSION 1 - REVIEW

REVIEWER	Shinichi Tanihara Department of Preventive Medicine and Public Health, School of Medicine, Fukuoka University, Japan
REVIEW RETURNED	12-Apr-2014

GENERAL COMMENTS	This study examined the association between self-reported eating rate and the prevalence of metabolic syndrome, using a large administrative database obtained from mandatory health check-ups for all employed workers. An epidemiologic study using extensive administrative data is important to assess health policy. However, I believe that the description of the database and the system of mandatory health check-ups in Japan is too vague for readers unfamiliar with the Japanese healthcare system to fully comprehend. For example, it is not mentioned that persons aged 40–74 are the target population of health check-ups for metabolic syndrome, as defined by the National Act on the Medical Care System for the Elderly Aged 75 and over. Major compulsory revisions: 1) A brief introduction to health check-ups as defined by Japanese law is needed to understand the contents of this paper, especially for foreign scientists.2) Because the professions of the study population are not mentioned in this article, you should be careful to generalize the study results. You should discuss this point as one of the limitations of this study.3) Because the study data is large-scale, parametric statistical methods should be used to analyse the data.
--

	4) The number of study participants for analysis should be corrected. Minor essential revisions: 1) 'Eating rate' : Please make sure this variable describes the speed/duration with which a person ingests their food. Otherwise, the correct term may be 'amount' (how much they eat) or 'frequency' (how often they eat) or another term. Next, you should add 'self-reported' because the eating rate was self-reported by participants. 2) 'Ministry of Health, Labor and Welfare' should be 'Ministry of Health, Labour and Welfare'. 3) 'normal continuous variables': Do you mean 'normal distribution'? 4) 'non-normal distribution': You should consider other types of distributions (for example, log-normal distribution). 5) You should describe a go of sake more precisely. For example, a go is a unit for liquid measurement and it is about 180 ml. Usually, Japanese sake contains about 14% ethanol and it is assumed that a go of sake contains about 25 g ethanol. 6) You should avoid simply repeating the same results expressed in the tables [for example, 1.00 (reference)].
--	---

REVIEWER	Sandra Roberta G. Ferreira School of Public Health, University of Sao Paulo, Brazil
REVIEW RETURNED	14-Apr-2014

GENERAL COMMENTS	 - Authors proposed to examine the association between eating rate and the presence (not "prevalence") of metabolic syndrome (MetS) in attendants of annual health examination in Japan. Additionally, frequencies of the MetS (and its components) according to sex and diagnostic criteria (NCEP and IDF), were presented in a large sample. - Ethics approval and participant consent are not mentioned. - In 2009, IDF and other experts agreed that waist circumference should not be an obligatory component to diagnose the MetS anymore. Three abnormal findings out of 5 would qualify a person for the MetS. They suggested national or regional cut points for waist circumference can be used. Therefore, this Reviewer would recommend to show prevalence data based only on their "joint interim statement". - It is not clear why authors decided to exclude individuals with history of myocardial infarction, coronary heart disease or stroke. - Assessment of "eating rate" was very simple. Please, consider provide details on duration for categories (slow, normal and fast), based on which (any) meal, expected degree of reliability etc. - When referring to multiple logistic regression data (Methods, Results and Discussion sections), authors should replace "prevalence" (of MetS) by "presence".
--

	- Some results concerning associations with MetS components are somehow unexpected. Please, elaborate on these. Table 2: once authors accept recommendation of using only "joint interim statement" to define MetS, this table should be simplified.
--	--

VERSION 1 – AUTHOR RESPONSE

Reviewer: 1

Reviewer Name Shinichi Tanihara

This study examined the association between self-reported eating rate and the prevalence of metabolic syndrome, using a large administrative database obtained from mandatory health check-ups for all employed workers. An epidemiologic study using extensive administrative data is important to assess health policy. However, I believe that the description of the database and the system of mandatory health check-ups in Japan is too vague for readers unfamiliar with the Japanese healthcare system to fully comprehend. For example, it is not mentioned that persons aged 40–74 are the target population of health check-ups for metabolic syndrome, as defined by the National Act on the Medical Care System for the Elderly Aged 75 and over.

Thank you very much for your thoughtful suggestions to our manuscript. According to your comments, we revised the manuscript as follows.

Major compulsory revisions:

1) A brief introduction to health check-ups as defined by Japanese law is needed to understand the contents of this paper, especially for foreign scientists.

Response:

We added the following sentence to the method and procedures section. "In Japan, health checkup under occupational health and safety law is mandatory for all employed workers and has been modified in 2008 when the recommendation for new national health checkup system focusing on MetS has been launched." (line 82-84 on page 6)

2) Because the professions of the study population are not mentioned in this article, you should be careful to generalize the study results. You should discuss this point as one of the limitations of this study.

Response

We added the point in the limitation. "The study participants were mainly workers in various industries including manufacturing (43.6%), service (27.8%) and transport and telecommunications (9.9%), and these figures are similar to those of national survey.⁴¹ However, information on profession of participants was not available, and thus caution is required when generalize the present finding." (line 240-244 on page 15)

3) Because the study data is large-scale, parametric statistical methods should be used to analyse the data.

Response:

Thank you for your suggestion. But we used non-parametric statistical methods, an extension of the Wilcoxon Rank-Sum test, to assess trend associations of triglyceride and fasting plasma glucose with eating rate in Table 1. Because the two values were skewed to the left. (line 133-137 on page 9, Table1)

4) The number of study participants for analysis should be corrected.

Response:

We confirmed that the numbers of participants were corrected in the previous version. We described the number of excluded participants in detail. And we already described that some participants met more than one of the exclusion criteria. (line 88-95 on page 7)

Minor essential revisions:

1) 'Eating rate' : Please make sure this variable describes the speed/duration with which a person ingests their food. Otherwise, the correct term may be 'amount' (how much they eat) or 'frequency' (how often they eat) or another term. Next, you should add 'self-reported' because the eating rate was self-reported by participants.

Response:

We already described that eating rate was assessed by asking "How fast is your speed of eating?" (line 104-106 on page 8). We added 'self-reported' to eating rate. (line 1 on page 1, line 21 on page 2, line 75 on page 6 and line 248 on page 17)

2) 'Ministry of Health, Labor and Welfare' should be 'Ministry of Health, Labour and Welfare'.

Response:

Thank you very much for pointing out the mistake. I corrected the term. (line 103 on page 7)

3) 'normal continuous variables': Do you mean 'normal distribution'?

Response:

I appreciate for your comments. As you mentioned, we used 'normal continuous variables' as 'continuous variables with normal distribution'. We deleted the term and revised sentence (line 133-135 on page 9)

4) 'non-normal distribution': You should consider other types of distributions (for example, log-normal distribution).

Response:

As you mentioned, we revised 'non-normal distribution' to 'skewed distribution'. (line 134 on page 9)

5) You should describe a go of sake more precisely. For example, a go is a unit for liquid measurement and it is about 180 ml. Usually, Japanese sake contains about 14% ethanol and it is assumed that a go of sake contains about 25 g ethanol.

Response:

Unfortunately, we don't have precise information on one go in our questionnaire. Then, we added the following sentence. "One go of sake, Japanese traditional beverage, is about 180 ml of 10 to 14% of ethanol, and contains ~23g of ethanol". (line 144 -145 on page 10)

6) You should avoid simply repeating the same results expressed in the tables [for example, 1.00 (reference)].

Response:

According to your suggestion, we modified tables; we deleted the term of '(reference)' and revised the footnote in Table2 and 3.

Reviewer: 2

Reviewer Name Sandra Roberta G. Ferreira

- Authors proposed to examine the association between eating rate and the presence (not "prevalence") of metabolic syndrome (MetS) in attendants of annual health examination in Japan. Additionally, frequencies of the MetS (and its components) according to sex and diagnostic criteria (NCEP and IDF), were presented in a large sample.

Thank you very much for your thoughtful suggestions to our manuscript. According to your comments, we revised the manuscript as follows.

- Ethics approval and participant consent are not mentioned.

Response:

We already mentioned Ethics approval in acknowledgment. But we didn't describe participant consent. So, in revised version we added participant consent and Ethics approval in methods sentence. (line 95-100 on page 7)

- In 2009, IDF and other experts agreed that waist circumference should not be an obligatory component to diagnose the MetS anymore. Three abnormal findings out of 5 would qualify a person for the MetS. They suggested national or regional cut points for waist circumference can be used. Therefore, this Reviewer would recommend to show prevalence data based only on their "joint interim statement".

Response:

Thank you very much for your thoughtful comments to our manuscript. According to your suggestion, we used "joint interim statement" to define MetS, instead of IDF and NCEP-ATP III criteria. And we revised our manuscript.

- It is not clear why authors decided to exclude individuals with history of myocardial infarction, coronary heart disease or stroke.

Response:

We excluded individuals with history of myocardial infarction, coronary heart disease or stroke because these severe diseases would affect both eating rate and metabolic syndrome.

- Assessment of "eating rate" was very simple. Please, consider provide details on duration for categories (slow, normal and fast), based on which (any) meal, expected degree of reliability etc.

Response:

We used a self-administered questionnaire to assess eating rate by asking "How fast is your speed of eating?", with three response options (slow, normal and fast). (line 105-107 on page 8) The questionnaire, which was recommended by the Japanese government (Ministry of Health, Labour and Welfare), was very common in Japanese health checkup system. Previous studies also used this questionnaire. Although the questionnaire was not validated yet, self-reported eating rate with five categories has been shown to be well correlated with friend-reported one, or objectively measured one. (line 234-235 on page 14)

- When referring to multiple logistic regression data (Methods, Results and Discussion sections), authors should replace "prevalence" (of MetS) by "presence".

Response:

Thank you very much for your suggestion. We replace "prevalence" by "presence". (line 42 on page 3, line 75 on page 6, line 139 on page 9, line 159, 160 and 168 on page 11, line 182 on page 12 and line 249 on page 16)

- Some results concerning associations with MetS components are somehow unexpected. Please, elaborate on these.

Response:

Thank you for your suggestion. We added the sentence of the association between eating rate and MetS components after adjustment for BMI. (line 199-209 on page 13)

Table 2: once authors accept recommendation of using only "joint interim statement" to define MetS, this table should be simplified.

Response:

According to your suggestion, we modified table 2, using "joint interim statement" to define metabolic syndrome.

VERSION 2 – REVIEW

REVIEWER	Shinichi Tanihara Department of Preventive Medicine and Public Health, School of Medicine, Fukuoka University, Japan
REVIEW RETURNED	17-Jun-2014

GENERAL COMMENTS	Major compulsory revisions Line 82-100 The number of attendants of 2011 (calendar year) annual health check examination at All Japan labor Welfare Foundation is 297,148 and about 70% (204,423) missed data for any of the components of metabolic syndrome (MetS). This is too high if the attendants focused on MetS. Please explain this. Line 134-137 You should consider other types of distributions (for example, log-normal distribution) for the "skewed distribution" and parametric statistical methods should be used to analyze the data. Minor essential revisions: Please check the amount of ethanol that a go of sake contains. Line 145: 23g of ethanol
---

REVIEWER	Sandra Roberta G. Ferreira School of Public Health, University of Sao Paulo, Brazil
REVIEW RETURNED	13-Jun-2014

GENERAL COMMENTS	This paper examined the association of self-reported eating rate with the presence of metabolic syndrome (MetS) in a large sample of Japanese workers including "some" dependents. The research question is simple and clear but the main variable (eating rate) is not objectively measured (only self-reported). The called "fast-food" - which is served quickly - is recognized as an energy-dense meal and usually is consumed at fast rate. Could the association of MetS with eating rate be biased to the high energy content of fast-foods? It is not clear why 3,660 individuals with history of cardiovascular diseases were excluded. The chance of having MetS should be high. In many countries, the absence of written informed consent from each participant would be a concern, but is not in Japanese Universities. Self-administered questionnaires used to assess the main variable (eating rate) is provided in Japanese; however, authors included some translated questions into English. Table 1: results concerning alcohol drinkers deserve some comment to facilitate interpretation. Findings reported in the Abstract and Results section are sometimes confusing; Table 3 is difficult to follow. For instance, lines 39-40 in the abstract: is really SLOW eating associated with high blood pressure and hyperglycemia? Lower eating rate seems to be associated with "protective" effects according to Table 3, but statistical significance disappears after adjustment for BMI. Footnotes from Table 3 deserve improvement because adjustments in the models vary according to the variable (component of MetS)
--

	under analysis. Discussion section, lines 183-185: Is it correct that "relationship with BP in both sexes and plasma glucose + lipids in men remained significant after adjustment for BMI? Please see Models 3g! There are several limitations but they are well discussed.
--	---

VERSION 2 – AUTHOR RESPONSE

Reviewer: 1

Reviewer Name Shinichi Tanihara

Major compulsory revisions

Line 82-100

The number of attendants of 2011 (calendar year) annual health check examination at All Japan labor Welfare Foundation is 297,148 and about 70% (204,423) missed data for any of the components of metabolic syndrome (MetS). This is too high if the attendants focused on MetS. Please explain this.

Response:

The majority of missing data for MetS components was plasma glucose (194,095 out of 204,423 participants). The major reason for a large number of participants with missing measurement of blood glucose is that they check HbA1c instead of blood glucose (139,125 out of 194,095 participants) because HbA1c substitutes for blood glucose under the health and safety law in Japan. Another reason is that young participant check neither blood glucose nor HbA1c (54,970 out of 194,095 participants), which was not required for whom aged under 40 except 35 year by the law in Japan. We elucidated the association eating rate and MetS precisely according to the JIS criteria, so a large number of participants were excluded from the present analysis.

In the manuscript, we revised to express the number of excluded participants sequentially. After exclusion of those who with cardiovascular disease and without information on eating rate, we excluded 182,487 participants with missing data for any of the components of MetS. Of 182,487 participants, 173,376 missed the data of blood glucose. (line 99-103 on page 8)

We added a possibility of bias due to such selective inclusion as a limitation. (line 260-262 on page 17)

You should consider other types of distributions (for example, log-normal distribution) for the "skewed distribution" and parametric statistical methods should be used to analyze the data.

Response:

Thank you for your advice. Fasting plasma glucose and triglyceride were highly skewed; hence they were log-transformed and expressed as geometric means (95% CI). (line 143-145 on page 11)

Minor essential revisions:

Please check the amount of ethanol that a go of sake contains.

Line 145: 23g of ethanol

Response:

Thank you for your thoughtful comment. We didn't define how much ethanol one go contains. We selected ~23g of ethanol according to previous studies. (line 155 on page 11)

Ref.

Urea Nitrogen Concentrations in Spot Urine, Estimated Protein Intake and Blood Pressure Levels in a Japanese General Population. *Am J Hypertension*, 23: 852-858.

Visceral Fat Area and Markers of Insulin Resistance in Relation to Colorectal Neoplasia. *Diabetes Care*, 33:184–189.

Alcohol Intake and Future Incidence of Hypertension in a General Japanese Population: The Hisayama Study. *Alcohol Clin Exp Res*, 26: 1010–1016.

Reviewer: 2

Reviewer Name Sandra Roberta G. Ferreira

This paper examined the association of self-reported eating rate with the presence of metabolic syndrome (MetS) in a large sample of Japanese workers including "some" dependents. The research question is simple and clear but the main variable (eating rate) is not objectively measured (only self-reported).

The called "fast-food" - which is served quickly - is recognized as an energy-dense meal and usually is consumed at fast rate. Could the association of MetS with eating rate be biased to the high energy content of fast-foods?

Response:

Thank you for your advice.

Some studies showed that fast-food, an energy-dense dietary source, was associated with MetS. But information on fast-foods was not available in our study. We cannot exclude the possibility of bias on the association of eating rate with MetS. We added the point to limitation section. (line 253-255 on page 17)

It is not clear why 3,660 individuals with history of cardiovascular diseases were excluded. The chance of having MetS should be high.

Response:

We excluded participants with cardiovascular disease because the disease effects on both eating rate and MetS. (line 97-99 on page 8)

In many countries, the absence of written informed consent from each participant would be a concern, but is not in Japanese Universities.

Response:

In Japan, informed consent is not necessarily required for observational studies using existing data, as described in the Japanese Ethical Guidelines for Epidemiological Research. We added that sentence to methods section. (line 106-108 on page 9)

Ref.

The Ministry of Health, Labour and Welfare, Ministry of Education, Culture, Sports, Science and Technology. Ethical guidelines for epidemiological research. Available at: <http://www.niph.go.jp/wadai/ekigakurinri/guidelines.pdf>. Accessed July 28, 2014.

Self-administered questionnaires used to assess the main variable (eating rate) is provided in Japanese; however, authors included some translated questions into English.

Response:

We translated the Japanese questionnaire which recommended by Ministry of Health, Labour and Welfare into English because we don't have the official English version of the questionnaire. We followed previous studies.

Ref .

Otsuka R, Tamakoshi K, Yatsuya H, et al. Eating fast leads to insulin resistance: findings in middle-aged Japanese men and women. *Prev Med* 2008;46:154-59.

Ohkuma T, Fujii H, Iwase M, et al. Impact of eating rate on obesity and cardiovascular risk factors according to glucose tolerance status: the Fukuoka Diabetes Registry and the Hisayama Study. *Diabetologia* 2013;56:70-77.

Sakurai M, Nakamura K, Miura K, et al. Self-reported speed of eating and 7-year risk of type 2 diabetes mellitus in middle-aged Japanese men. *Metabolism* 2012;61:1566-71.

Table 1: results concerning alcohol drinkers deserve some comment to facilitate interpretation.

Response:

Thank you for your advice. Both men and women who ate fast consumed greater amount of alcohol than those who ate slowly. We added the description in the results section. (line 166-167 on page 12)

Findings reported in the Abstract and Results section are sometimes confusing; Table 3 is difficult to follow. For instance, lines 39-40 in the abstract: is really SLOW eating associated with high blood pressure and hyperglycemia? Lower eating rate seems to be associated with "protective" effects according to Table 3, but statistical significance disappears after adjustment for BMI.

Response:

Thank you for your pointing out our mistake. We corrected the wrong sentence in abstract and result. After adjustment for body mass index, the significant trend association disappeared but slow eating was associated with decreased odds of high blood pressure (men and women) and hyperglycemia (men) and fast eating was associated with increased odds of lipid abnormality (men). (line 39-41 on page 4 and line 188-190 on page 13)

Footnotes from Table 3 deserve improvement because adjustments in the models vary according to the variable (component of MetS) under analysis.

Response:

We used the same model according to the variables. We found mistake of one superscript and corrected it.

Discussion section, lines 183-185: Is it correct that "relationship with BP in both sexes and plasma glucose + lipids in men remained significant after adjustment for BMI? Please see Models 3g!

Response:

Thank you for your suggestion. We revised the sentence as follows. "Even after adjustment for BMI, slow eating was associated with decreased odds of high blood pressure in both men and women and high fasting plasma glucose in men, and fast eating was associated with increased odds of lipid abnormal profiles in men, compared to normal eating rate." (line 195-198 on page 14)

There are several limitations but they are well discussed.

Your words are encouraging for us.

VERSION 3 - REVIEW

REVIEWER	Sandra Roberta G. Ferreira University of Sao Paulo School of Public Health Brazil
REVIEW RETURNED	07-Aug-2014

GENERAL COMMENTS	Authors have answered appropriately to the majority of the criticisms. Unformatted layout of the Tables makes difficult to follow their content and, consequently, the reading of the Results chapter.
--